# A Universal Bleeding Risk Score in Native and Allograft Kidney Biopsies: A French Nationwide Cohort Study

**DOI:** 10.3390/jcm12103527

**Published:** 2023-05-17

**Authors:** Mathieu Kaczmarek, Jean-Michel Halimi, Jean-Baptiste de Fréminville, Philippe Gatault, Juliette Gueguen, Nicolas Goin, Hélène Longuet, Christelle Barbet, Arnaud Bisson, Bénédicte Sautenet, Julien Herbert, Matthias Buchler, Laurent Fauchier

**Affiliations:** 1Néphrologie-Immunologie Clinique, Hôpital Bretonneau, CHU Tours, F-37000 Tours, France; mathieu.kaczmarek@univ-tours.fr (M.K.); philippe.gatault@univ-tours.fr (P.G.); j.gueguen@chu-tours.fr (J.G.); ni.goin@gmail.com (N.G.); h.longuet@chu-tours.fr (H.L.); c.barbet@chu-tours.fr (C.B.); benedicte.sautenet@gmail.com (B.S.); mathias.buchler@univ-tours.fr (M.B.); 2EA4245, University of Tours, F-37000 Tours, France; 3INI-CRCT, F-54500 Nancy, France; 4Paris-Cardiovascular Research Center, INSERM, UMR970, Université de Paris, F-75006 Paris, France; jeanbaptiste.defreminville@aphp.fr; 5Unité Fonctionnelle d’Hypertension Artérielle, Centre de Référence des Maladies Rares de la Surrénale, Hôpital Européen Georges Pompidou, Assistance Publique-Hôpitaux de Paris, F-75015 Paris, France; 6Service de Cardiologie, Centre Hospitalier Universitaire Trousseau et Faculté de Médecine, EA7505, Université de Tours, F-37000 Tours, France; arnaud.bisson37@gmail.com (A.B.); j.herbert@chu-tours.fr (J.H.); laurent.fauchier@univ-tours.fr (L.F.); 7Service d’Information Médicale, d’Épidémiologie et d’Économie de la Santé, Centre Hospitalier Universitaire et Faculté de Médecine, EA7505, Université de Tours, F-37000 Tours, France

**Keywords:** biopsy, bleeding, score, native kidney, kidney graft, epidemiology

## Abstract

Background: The risk of bleeding after percutaneous biopsy in kidney transplant recipients is usually low but may vary. A pre-procedure bleeding risk score in this population is lacking. Methods: We assessed the major bleeding rate (transfusion, angiographic intervention, nephrectomy, hemorrhage/hematoma) at 8 days in 28,034 kidney transplant recipients with a kidney biopsy during the 2010–2019 period in France and compared them to 55,026 patients with a native kidney biopsy as controls. Results: The rate of major bleeding was low (angiographic intervention: 0.2%, hemorrhage/hematoma: 0.4%, nephrectomy: 0.02%, blood transfusion: 4.0%). A new bleeding risk score was developed (anemia = 1, female gender = 1, heart failure = 1, acute kidney failure = 2 points). The rate of bleeding varied: 1.6%, 2.9%, 3.7%, 6.0%, 8.0%, and 9.2% for scores 0 to 5, respectively, in kidney transplant recipients. The ROC AUC was 0.649 (0.634–0.664) in kidney transplant recipients and 0.755 (0.746–0.763) in patients who had a native kidney biopsy (rate of bleeding: from 1.2% for score = 0 to 19.2% for score = 5). Conclusions: The risk of major bleeding is low in most patients but indeed variable. A new universal risk score can be helpful to guide the decision concerning kidney biopsy and the choice of inpatient vs. outpatient procedure both in native and allograft kidney recipients.

## 1. Introduction

Transplant kidney biopsy is an important tool for kidney transplant management [1]. It is the gold standard for identifying the cause of kidney graft dysfunction in many cases [2]. Protocol transplant kidney biopsies are performed in many teams, even for patients with stable renal function during the first months after transplantation, to identify subclinical rejections, calcineurin inhibitor toxicity and allograft nephropathy [3]. Since the first biopsy in 1953 [4], the complication rate and diagnostic yield have been improved by technical refinement (echographic guidance [5,6], automated trocar [7], thinner needle [8], kidney transplant approach [9]), making kidney transplant biopsy a potential outpatient procedure [10,11].

However, kidney transplant biopsy remains one of the echo-guided procedures with the highest bleeding risk according to the International Society of Radiology [12]. Rates of bleeding complications are highly variable, from 1.8% to 14% [13,14,15,16,17,18,19]. Older studies reported a high rate of complications, but even in recent studies, the rate of bleeding in kidney transplant biopsies remained variable [13,20,21,22,23,24]. Death or kidney transplant loss following biopsy are rare (from 0% to 0.3% and from 0% to 0.6%, respectively) [13,14,15,16,17,18,19]. Nevertheless, the problem of blood transfusion should not be dismissed, as it is a major risk factor for allo-sensitization and therefore remains a concern in this population [25]. Only a few bleeding risk factors have been identified [14,17,21,22,23]. The better identification of the risk factors for bleeding after kidney transplant biopsy is crucial to building bleeding risk scores, but unfortunately, no bleeding risk score has been developed for kidney transplant recipients.

In the present paper, using nationwide data, we assessed the rate and risk factors associated with major bleeding complications after percutaneous kidney transplant biopsies from 2010 to 2019 in kidney transplant recipients [24,26]. We proposed a new bleeding risk score in this population and assessed whether this new score could be adapted for patients with a percutaneous native who had a kidney biopsy during the same period.

## 2. Materials and Methods

### 2.1. Study Design

This longitudinal cohort study was built on the national hospitalization database covering hospital care for the entire French population. Data of patients admitted to hospital for a native or transplant kidney biopsy in France between January 2008 and December 2019 were collected from the national medico-administrative “programme de médicalisation des systèmes d’information” (PMSI) database (i.e., medicalized information system program). Shortly, obligatory since 1991 in public and private structures, this database covers more than 98% of the French population (67 million people) from birth (or immigration) to death (or emigration). Medical information, including principal and secondary diagnoses, and procedures are respectively recorded anonymously according to the International Classification of Diseases, tenth revision (ICD-10) and the “Classification Commune des Actes Médicaux” (CCAM). This database was already used in various works of different medical specialties and was reliable [24,26]. All the data were anonymized, so ethical approval was not required. The French Data Protection Authority granted access to the PMSI data. The procedures for data collection and management were approved by the Commission Nationale de l’Informatique et des Libertés (CNIL), the independent National Ethical Committee protecting human rights in France, which ensures that all information is kept confidential and anonymous in compliance with the Declaration of Helsinki. This study required neither information nor non-opposition of the included individuals. Access to the linked anonymous file in the PMSI database was approved by the CNIL (MR-005 registration number 0415141119).

### 2.2. Patient Selection

Kidney transplant recipients who were admitted for a kidney biopsy during the 2010–2019 period were included in the present study. We also included as a control group nontransplanted patients who had a percutaneous native kidney biopsy during the same period (ICD-10 codes: JAHB001, JAHJ006, JAGH007, Z940).

### 2.3. Major Bleeding and Risk of Death after Biopsy

A major bleeding complication was defined by blood transfusion (ICD-10 code: FELF011), hematoma/hemorrhage (ICD-10 code: T810), angiographic intervention (ICD-10 code: EDSF003, EDSF008) or nephrectomy (ICD-10 code: JAFA002, JAFA023) occurring during an 8-day period following native or graft kidney biopsy and was determined by the administrative codes. For the risk of death associated with a major bleeding event after biopsy, a 30-day period was considered.

### 2.4. Collected Data

#### 2.4.1. Demographic, Cardiovascular and Metabolic Conditions

Patient information was extracted from the data collected in the hospital records. For each hospital stay, the diagnoses at discharge were obtained. Each variable was identified using the ICD-10 codes. We also used the Charlson Comorbidity Index [27] and the Claims-based Frailty Indicator [28] to assess patients’ clinical status. As the information was based on codes, there was no missing value. Conditions of interest included hypertension, diabetes, obesity, heart failure, valve diseases, coronary artery disease, smoking, dyslipidemia, stroke, vascular disease, atrial fibrillation and acute kidney failure.

#### 2.4.2. Other Relevant Parameters

We collected information regarding patients’ history of alcohol-related diagnoses, lung diseases, liver diseases, cancer within the years preceding the biopsy, thrombocytopenia and anemia. Of note, patients’ medications, including antiplatelet agents and anticoagulants, coagulation parameters, time between transplantation and biopsy and needle size were not available.

### 2.5. Statistical Analyses

Data are presented as means and standard deviations for quantitative parameters and percentages for categorical parameters. Patients who had major bleeding complications (blood transfusion, hematoma/hemorrhage, angiographic intervention or nephrectomy) during an 8-day period after biopsy were compared to other patients using Student’s t test or the Chi^2^ as appropriate. Multivariable logistic regressions were used, and the results were expressed as odd ratios (OR) and 95% confidence intervals (95% CI). Using the results of the multivariable logistic regressions, we built a new score for major bleeding after biopsy in kidney transplant recipients. To create the score points, the regression coefficients with *p* < 0.01 were divided by the smallest coefficient and rounded to the nearest integer, which assigned a given number of points to each significant predictor in the mortality model [29,30]. We also assessed the diagnostic performance of the previously validated bleeding score (developed for native kidneys) in this population of kidney transplant recipients [24,26].

As this score is easier to use than the one previously published and developed in patients with native kidneys, we assessed whether this simple new score could also be used in patients with percutaneous native kidney biopsies. 

Receiver operating characteristic (ROC) curves were constructed and the areas under the curve (AUC) with 95% confidence intervals for this score were compared using the DeLong test. 

## 3. Results

### 3.1. Baseline Characteristics

In the present study, 28,034 patients with kidney transplant biopsy were included (Table 1). The mean age was 51.6 ± 14.7 years, and two-thirds of patients were male. Hypertension and diabetes were present in 74.8% and 26.7% of patients, respectively. Obesity was noted in 13.8% of patients. Congestive heart failure (13.6%), anemia (51.7%), and acute renal failure (48.7%) were also present in many patients.

We used the data of 55,026 patients who had native percutaneous kidney biopsies within the same period as a control, as these two populations were not comparable (Table 1).

### 3.2. Major Bleeding Rate in Transplant Kidney Biopsies

The major bleeding rate was low in this population, with 56/28,034 (0.2%) angiographic interventions, 104/28,034 (0.4%) hemorrhage/hematomas, 7/28,034 (0.02%) nephrectomies and 1238/28,034 (4.0%) blood transfusions (Table 2). Expectedly, the rate of major bleeding was lower in this population than in patients with native kidney biopsies (Table 2) (adjusted OR: 0.53 [95% CI, 0.48–0.57], *p* < 0.0001).

The rate of death at day 30 was 0.11% (32/28,034) in the kidney transplant recipients (as a control, the rate was 543/55,026 (1.0%) in the patients with native kidney biopsies) (Table 2).

### 3.3. Risk Factors for Major Bleeding and Development of a New Score in This Population

Female gender, heart failure, anemia, and acute renal failure were significant risk factors for major bleeding in kidney transplant recipients (Table 3). 

Using these parameters, we built a new risk score from 0 to 5 points (female gender = 1 point, heart failure = 1 point, anemia = 1 point, acute renal failure = 2 points) (Table 3). The rate of bleeding varied from 1.6% to 9.2% from the lowest to the highest risk groups (Figure 1). 

The AUC ROC curve associated with this score was 0.649 (99%CI: 0.635–0.664) (Figure 2a). When major bleeding was defined as transfusion, angiographic intervention or nephrectomy (but not hematoma/hemorrhage), the AUC was 0.657 (0.642–0.672) (Figure 2b).

### 3.4. Application of This New Score to Patients with Native Kidneys

Using the new score in patients with native kidney biopsies, the AUC was 0.755 (0.746–0.763) and 0.777 (0.769–0.785), respectively, according to whether or not hematoma/hemorrhage was included in the definition of major bleeding, indicating that this new score can be used for native kidneys. The rate of bleeding varied from 1.2% (score = 0) to 19.2% (score = 5) (Appendix A).

### 3.5. Major Bleeding Risk: Center Effect in Kidney Transplant Recipients

We assessed the effect of the percutaneous biopsy procedure frequency using the quartile of volume on bleeding events for centers with more than 10 biopsies over 2010 to 2019. A significant center effect was found for percutaneous kidney transplant biopsies (OR 0.93 [95% CI, 0.88–0.99]) (Table 4).

## 4. Discussion

In the present study, we assessed the risk of major bleeding in 28,034 consecutive kidney transplant recipients who had a kidney biopsy during the 2008–2019 period. The risk of major bleeding was low in many patients (angiographic intervention: 0.2%, hemorrhage/hematoma: 104/28,034: 0.4%, nephrectomy: 0.02%, blood transfusion: 4.0%), although it could be as high as 9.2% in some patients. Anemia, female gender, heart failure and acute kidney failure were identified as risk factors for major bleeding. We were able to propose a new major bleeding risk score, which was initially proposed for kidney transplant recipients but which can be used also in native kidneys.

First, percutaneous kidney transplant biopsy is a relatively safe procedure with a low risk of major bleeding complications. The rate of complications in the literature varies from 0% to 2.8% for protocol biopsies [13,16,17,31,32] to 0% to 3.4% for cause biopsies [13,16,17,24]. However, in a recent large nationwide sample of 14,268 percutaneous kidney transplant biopsies, a higher rate of bleeding complications was reported (blood transfusion: 4.9%, angiographic intervention: 0.4%) [22]. These rate differences could be explained by the higher rate of blood transfusion in our study. Indeed, the rates of angiographic procedure and nephrectomy were similar (0–0.3% and 0–0.6%, respectively), although the rate of blood transfusion was lower in other reports (0–3.3%) [13,14,15,16,17]. These reports were usually small and monocentric, and selection bias may be present. The rate differences could also be explained by publication biases in the literature. 

Second, the risk of bleeding was highly variable according to patient characteristics. We were able to identify risk factors for major bleeding, such as anemia, female gender and heart failure and acute kidney failure. Female gender has been identified as a risk factor for major bleeding in percutaneous and transjugular native kidney biopsies, and in kidney transplant recipients [24,25,26,33]. Renal dysfunction was also identified as a risk factor for bleeding in many, if not all, reports [14,17,32,34]. Whether anemia per se is a risk factor for bleeding after biopsy is debated in the literature [35], even if the association between anemia and bleeding was reported in a systematic review [36]. Patients with anemia before biopsy are more prone to receive blood transfusion after biopsy, even in the absence of demonstrated external hemorrhage or hematoma. To better assess the specific role of anemia as a risk factor for bleeding, it would have been interesting to perform a sensitivity analysis using nephrectomy, hemorrhage/hematoma and nephrectomy as bleeding events (excluding, therefore, blood transfusion). Unfortunately, blood transfusion represented 90.3% of all the bleeding events in the kidney transplant recipients (and 87.4% for the percutaneous native kidney biopsies), and therefore, such an analysis was not possible. Of note, a center volume effect on major bleeding complications was found in the kidney transplant recipients. This finding was also reported in a nationwide study in Norway [34]. The use of blood transfusion after bleeding may be different among centers, and different practices may explain the current findings.

Third, based on simple parameters known at the time of biopsy, we now propose a simple pre-procedure major bleeding risk score. Although the diagnostic performance of this new score was lower than our previous score validated for percutaneous or transjugular biopsies [24,26], it is similar to universally used scores such as the CHA_2_DS_2_-VASC score (used to predict the risk of ischemic stroke in patients with atrial fibrillation (AUC = 0.672)) and Geneva revised score (used to predict pulmonary embolism (AUC = 0.693)) [27,28]. Consequently, our score could be helpful in daily practice in kidney transplant recipients. The use of this score could be particularly important for protocol biopsies in kidney allografts: in patients with high risks identified with this score, it can be preferable not to perform biopsies. In kidney transplant recipients in whom biopsy is deemed necessary because of proteinuria or acute kidney injury, it may be preferable to prefer inpatient vs. outpatient biopsies in high-risk patients. We do not propose our previous score validated in patients with percutaneous and transjugular kidney biopsies because it was cumbersome, as this score required the calculation of the Charlson Index and Frailty Indicator score. Our new score had very good performance for percutaneous biopsies of native kidney biopsies (AUC: 0.755 [0.746–0.763]); therefore, it could be very helpful in real-life conditions, as only the four parameters needed for this score (anemia, female gender, acute kidney injury, heart failure) are readily available for these patients. 

We noticed that the rate of death at 30 days after biopsy in the kidney transplant recipients was 0.1%, in accordance with other reports, and it was much lower than in the patients with native kidneys. We observed an increased risk of death after biopsy in the patients with bleeding complications. However, it is impossible to ensure that death after biopsy was due to the complications of biopsy, and it is possible that death was more related to the underlying diseases of the patients. 

Our work has some limits. This study is based on administrative data obtained and manually filled by physicians and administrators. The data were not systematically externally checked, and this could have caused information bias. However, as coding is linked to reimbursement and is regularly controlled, it is expected to be of good quality. Our analysis based on the administrative codes was restricted to the variables present in the database. As indicated in the Section 2, demographics, comorbidities, medical history, and events during hospitalization or follow-up were collected from the hospital records using the ICD-10 codes, and as the information was based on codes, there was no missing value. Moreover, the role of some parameters, such as coagulation parameters, presence of antiplatelet agents or anticoagulants and their withdrawal before biopsy, size of gauge and number of passes, and specialty of operators (radiologist or nephrologist), could not be analyzed. Whether anemia per se is a risk factor for bleeding after biopsy is debated in the literature, even if the association between anemia and bleeding was reported in a systematic review [35,36]. Patients with anemia before biopsy are more prone to receive blood transfusion after biopsy, even in the absence of demonstrated external hemorrhage or hematoma. To better assess the specific role of anemia as a risk factor for bleeding, it would have been interesting to perform a sensitivity analysis using nephrectomy, hemorrhage/hematoma and nephrectomy as bleeding events (excluding, therefore, blood transfusion). Unfortunately, blood transfusion represented 90.3% of all the bleeding events in the kidney transplant recipients (and 87.4% for the percutaneous native kidney biopsies), and therefore, such an analysis was not possible.

The strengths of our work reside in its size and design. To the best of our knowledge, it represents the largest analysis focusing on the issue of major bleeding complications after percutaneous kidney transplant biopsy. Our findings are in accordance with another nationwide study [34]. Choosing an eight-day period after percutaneous biopsy allowed us to estimate the risk of complications. Therefore, our findings, based on real-life data, could be used in clinical practice.

## 5. Conclusions

In conclusion, the risk of major bleeding is low but highly variable according to risk factors such as thrombocytopenia, anemia, female gender, and heart failure. A new simple major bleeding risk score specific to kidney transplant recipients is now proposed, and it can be helpful to guide the decision concerning kidney biopsy (including the choice of no biopsy for protocol biopsies in patients with stable renal function when the risk of bleeding is considered excessive) and the choice of the most adequate procedure (inpatient vs. outpatient). It can also be a tool for patients and physicians to facilitate shared decision-making.

## Figures and Tables

**Figure 1 jcm-12-03527-f001:**
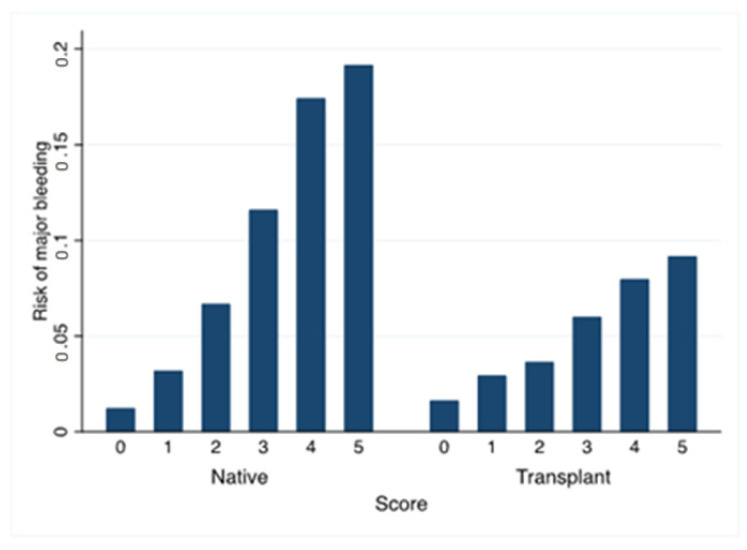
Risk of major bleeding according to the new major bleeding risk score. Risk of major bleeding (angiographic intervention, nephrectomy, blood transfusion, hemorrhage/hematoma) at day 8 in relation to the number of points for this score (from 0 to 5 points) in kidney transplant recipients and in patients with percutaneous native kidney biopsies. The proportion of major bleeding according to the score in the kidney transplant biopsies was 90/5517 (1.6%) for score = 0, 174/5931 (2.9%) for score = 1, 199/5459 (3.7%) for score = 2, 370/90/6160 (6.0%) for score = 3, 336/4215 (8.0%) for score = 4, and 69/752 (9.2%) for score = 5). The respective figures for the percutaneous native kidney biopsies were 237/19,358 (1.2%) for score = 0, 489/15,309 (3.2%) for score = 1, 552/8,263 (6.7%) for score = 2, 814/7021 (11.6%) for score = 3, 735/4219 (17.4%) for score = 4, and 164/856 (19.2%) for score = 5).

**Figure 2 jcm-12-03527-f002:**
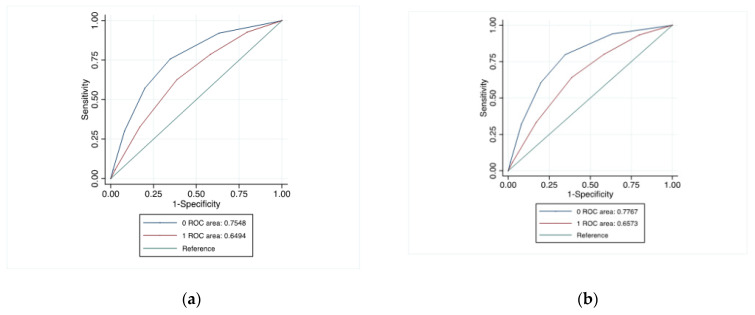
Major bleeding risk score ROC curve in kidney transplant recipients (red line) and native percutaneous kidney biopsy patients (blue line). (**a**) Major bleeding defined by angiographic intervention, nephrectomy, transfusion, hematoma/hemorrhage at day 8. (**b**) Major bleeding defined by angiographic intervention, nephrectomy and transfusion at day 8. The score for major bleeding initially developed for native kidney biopsies [24,26] was tested in kidney transplant recipients, and the diagnostic performance of this score was similar (AUC: 0.662 [95% CI, 0.648–0.676], Appendix A; the rate of bleeding from 0.2% (score: 0–4 points) to 15.1% (score ≥ 35 points), Appendix A).

**Table 1 jcm-12-03527-t001:** Baseline characteristics.

	Percutaneous Native Kidney Biopsy	Percutaneous Biopsy in Kidney Transplant
	**(*n* = 55,026)**	**(*n* = 28,034)**
**Age, years**	58.0 ± 17.4	51.6 ± 14.7
**Sex (male)**	33,523 (60.9)	17,706 (63.2)
**Charlson comorbidity index**	4.6 ± 2.8	4.8 ± 2.7
**Frailty index**	7.1 ± 7.7	8.6 ± 7.6
**Hypertension**	29,392 (53.4)	20,971 (74.8)
**Diabetes mellitus**	12,376 (22.5)	7476 (26.7)
**Obesity**	8626 (15.7)	3863 (13.8)
**Heart failure with congestion**	5827 (10.6)	3825 (13.6)
**Valve disease**	2454 (4.5)	1189 (4.2)
**Coronary artery disease**	5486 (10.0)	4404 (15.7)
**Vascular disease**	5889 (10.7)	4081 (14.6)
**Atrial fibrillation**	4839 (8.8)	2090 (7.5)
**Ischemic stroke**	1006 (1.8)	362 (1.3)
**Smoker**	5565 (10.1)	2726 (9.7)
**Dyslipidemia**	10,257 (18.6)	6440 (23.0)
**Poor nutrition**	4865 (8.8)	2077 (7.4)
**Alcohol related diagnoses**	3689 (6.7)	939 (3.3)
**Lung disease**	5749 (10.4)	2070 (7.4)
**Liver disease**	3345 (6.1)	1493 (5.3)
**Anaemia**	13,382 (24.3)	14,484 (51.7)
**Thrombocytopenia**	3854 (7.0)	1733 (6.2)
**Previous cancer**	13,264 (24.1)	2562 (9.1)
**Abnormal renal function**	17,566 (31.9)	23,791 (84.9)

**Table 2 jcm-12-03527-t002:** Major bleeding rate after kidney biopsy at day 8.

	Percutaneous Biopsy in Kidney Transplant	Percutaneous Native Kidney Biopsy
	**(*n* = 28,034)**	**(*n* = 55,026)**
**Angiographic intervention**	56 (0.2)	216 (0.4)
**Nephrectomy**	7 (0.0)	33 (0.1)
**Blood transfusion**	1118 (4.0)	2614 (4.8)
**Hemorrhage/hematoma**	104 (0.4)	273 (0.5)
**Any of the bleeding events**	1238 (4.4)	2991 (5.4)
**Angiographic intervention or nephrectomy or transfusion**	1160 (4.1)	2778 (5.0)
**Death at day 30**	32 (0.1)	543 (1.0)

**Table 3 jcm-12-03527-t003:** Risk factors for major bleeding in kidney transplant recipients.

	Univariate Analysis		Multivariable Analysis	
	HR, 95% CI	*p*	HR, 95% CI	*p*
**Age (quartile)**	0.980 (0.925–1.039)	0.49	0.907 (0.848–0.970)	0.004
**Charlson comorbidity index**	1.272 (1.206–1.341)	<0.0001	1.240 (1.155–1.333)	<0.0001
**Frailty index**	1.288 (1.219–1.361)	<0.0001	1.139 (1.072–1.211)	<0.0001
**Sex (male)**	0.797 (0.710–0.895)	<0.0001	0.853 (0.756–0.962)	0.01
**Hypertension**	1.138 (0.994–1.302)	0.06	0.844 (0.725–0.984)	0.03
**Diabetes mellitus**	1.088 (0.959–1.235)	0.19	0.848 (0.720–0.999)	0.05
**Heart failure with congestion**	1.750 (1.520–2.015)	<0.0001	1.194 (1.018–1.401)	0.03
**Valve disease**	1.397 (1.092–1.789)	0.008	1.048 (0.807–1.362)	0.72
**Coronary artery disease**	1.182 (1.018–1.371)	0.03	1.035 (0.868–1.235)	0.70
**Vascular disease**	1.074 (0.917–1.258)	0.37	0.805 (0.671–0.965)	0.02
**Atrial fibrillation**	1.473 (1.221–1.778)	<0.0001	1.170 (0.951–1.438)	0.14
**Ischemic stroke**	1.339 (0.858–2.088)	0.20	0.937 (0.594–1.478)	0.78
**Smoker**	1.194 (0.997–1.430)	0.06	0.993 (0.818–1.206)	0.95
**Dyslipidemia**	1.032 (0.902–1.180)	0.65	0.940 (0.807–1.096)	0.43
**Obesity**	1.164 (0.994–1.362)	0.06	1.035 (0.874–1.226)	0.69
**Poor nutrition**	1.470 (1.217–1.775)	<0.0001	0.958 (0.785–1.169)	0.67
**Alcohol related diagnoses**	1.335 (1.008–1.768)	0.04	1.000 (0.739–1.352)	1.00
**Abnormal renal function**	1.889 (1.548–2.306)	<0.0001	1.328 (1.070–1.648)	0.01
**Lung disease**	1.275 (1.045–1.555)	0.02	0.894 (0.725–1.103)	0.30
**Liver disease**	1.510 (1.217–1.874)	<0.0001	1.016 (0.804–1.285)	0.89
**Anaemia**	2.262 (1.998–2.560)	<0.0001	1.710 (1.490–1.963)	<0.0001
**Thrombocytopenia**	1.709 (1.408–2.074)	<0.0001	1.081 (0.879–1.329)	0.46
**Previous cancer**	1.091 (0.901–1.322)	0.37	0.855 (0.697–1.049)	0.13
**Acute renal failure**	2.499 (2.209–2.827)	<0.0001	1.870 (1.640–2.132)	<0.0001

**Table 4 jcm-12-03527-t004:** Variability of major bleeding rate after kidney biopsy according to center volume.

Quartile of Center Volume	Mean Number of Biopsies by Center, 2010–1019	Number of Patients with Percutaneous Biopsies, *n*	Number of Patients with Native Biopsy/Transplant	Major Bleeding Among Patients with Percutaneous Biopsy, *n* (%)	Major Bleeding among Patients with Native Percutaneous Biopsy, *n* (%)	Major Bleeding Among Patients with Transplant Percutaneous Biopsy, *n* (%)
**1**	159 ± 86	20,827	17,936/2891	1081 (5.2)	949 (5.3)	132 (4.6)
**2**	524 ± 121	21,661	15,835/5826	1268 (5.9)	956 (6.0)	312 (5.4)
**3**	1035 ± 235	22,778	12,425/10,353	1063 (4.7)	653 (5.3)	410 (4.0)
**4**	4473 ± 1121	17,794	8830/8964	817 (4.6)	433 (4.9)	384 (4.3)

Volume was evaluated for centers with more than 10 biopsies over 2010–2019. Odds ratio for the risk of major bleeding based on center volume in patients with native percutaneous biopsy = 0.97 (95% CI 0.94–1.01), *p* = 0.11. Odds ratio for the risk of major bleeding based on center volume in patients with transplant percutaneous biopsy = 0.93 (95% CI 0.88–0.99), *p* = 0.02.

## Data Availability

Because of the sensitive nature of the data collected for this study, requests to access the dataset from qualified researchers trained in human subject confidentiality protocols may be sent to the Institut National des Données de Santé (http://www.indsante.fr/).

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
