# Peer review of "A Universal Bleeding Risk Score in Native and Allograft Kidney Biopsies: A French Nationwide Cohort Study"

_jcm, 2023, doi:10.3390/jcm12103527_

Round 1

Reviewer 1 Report

I enjoyed reading your article and found it interesting. 

As you have mentioned, it is unfortunate that coagulation and medication information was not available.

Could you please clarify how accurately the coding system captures data in France? Meaning, is the coding reliable or there is a good chance of data missing? Perhaps you can make a comment on this.

Also, what are your thoughts about mortality after the biopsy? Are you suggesting the mortality was related somehow to the biopsy?  Could this not be the result of the underlying disease?

Author Response

1.As you have mentioned, it is unfortunate that coagulation and medication information was not available.

We have acknowledged this limit in the Discussion section. However, aspirin and anticoagulant medications are usually withdrawn before kidney biopsy in most centers, soi t is not a real issue in most situations.

2.Could you please clarify how accurately the coding system captures data in France? Meaning, is the coding reliable or there is a good chance of data missing? Perhaps you can make a comment on this.

As indicated in the Methods section, demographics, comorbidities, medical history, and events during hospitalization or follow-up are collected from the hospital records using ICD-10 codes, and as the information was based on codes, there was no missing value.

We propose to add in the Discussion section :

« This study is based on administrative data obtained and manually filled by physicians and administrators. Data were not systematically externally checked, and this could have caused information bias. However, as coding is linked to reimbursement and is regularly controlled, it is expected to be of good quality. Our analysis based on administrative codes was restricted to the variables present in the database. As indicated in the Methods section, demographics, comorbidities, medical history, and events during hospitalization or follow-up are collected from the hospital records using ICD-10 codes, and as the information was based on codes, there was no missing value.”

3.what are your thoughts about mortality after the biopsy? Are you suggesting the mortality was related somehow to the biopsy?  Could this not be the result of the underlying disease?

We understand your point : it is impossible to ensure that death after biopsy was due to the complications of biopsy, and it is possible that death was more related to the underlying diseases of the patients.

We propose to add in the Discussion section :

« We observed an increased risk of death after biopsy in patients with bleeding complications. However, it is impossible to ensure that death after biopsy was due to the complications of biopsy, and it is possible that death was more related to the underlying diseases of the patients. »

Reviewer 2 Report

Dear editor,

I have read the paper with interest. Authors investigates a clinically relevant topic. The manuscripts reads well. However, a few points deserve mention:

  • the statistical analysis session should be expanded and it should describe in details how were MV adjusted analysis carried out
  • Is anaemia a real risk factors for bleeding or is a proxy of blood transfusion (which is used to define the endpoint of bleeding)? A sensitivity analysis is suggested to investigate whether anemia is a predictor of the endpoint defined whiteout the code for blood transfusion. 
  • Similarly, authors describe a significant center effect. However, the use of blood transfusion after bleeding maybe different among centres and different practices may explain current findings
  • If the sensitivity analysis would suggest that anemia is the real driver of the association with the endpoint as defined by authors, the reviewer would suggest changing the title to suggest the need for blood transfusion and the discussion to highlight that the focus is not the entity of the bleeding per se but the need for blood transfusion the scope of the analysis. Ultimately, blood transfusion is what w should avoid to reduce the risk of allo-sentitization 
  • Please add in the text a Breuer explanation of how the new bleeding risk is calculated. Also explain how the AUC analysis are carried out (bleeding risk as a continuous variable? What is the cut off? As dichotomous variable high vs low risk (what is the definition of high vs low risk)?
  • Can authors compare the additive prognostic value of the new score when compared to other risks scores?

Author Response

1.The statistical analysis session should be expanded and it should describe in details how were MV adjusted analysis carried out.

2.Is anaemia a real risk factors for bleeding or is a proxy of blood transfusion (which is used to define the endpoint of bleeding)?

This is a real issue in the literature.

Patients with anemia before biopsy are more prone to receive blood transfusion after biopsy, even in the absence of demonstrated external hemorrhage or hematoma. On the other hand, some minor bleedings were certainly missed. In France, systematic post-biopsy imaging is not universally done, especially when the fall in hemoglobin is minor or absent and when no pain is present. However, anemia per se as a risk of bleeding was also highlighted in the systemic review of Corapi et al (Corapi KM, Chen JL, Balk EM, Gordon CE. Bleeding complications of native kidney biopsy: a systematic review and meta-analysis. Am J Kidney Dis 2012; 60:62–73).

We propose to add in the Discussion section :

Whether anemia per se is a risk factor for bleeding after biopsy is debated in the literature (Ref 34 : Halimi et al, Curr Op Nephrol Hypertens 2021), even if the association between anemia and bleeding was reported in a systematic review (Corapi et al (Corapi KM, Chen JL, Balk EM, Gordon CE. Bleeding complications of native kidney biopsy: a systematic review and meta-analysis. Am J Kidney Dis 2012; 60:62–73). Patients with anemia before biopsy are more prone to receive blood transfusion after biopsy, even in the absence of demonstrated external hemorrhage or hematoma.

3.A sensitivity analysis is suggested to investigate whether anemia is a predictor of the endpoint defined whitout the code for blood transfusion.

We understand your point, and it is valid.

As shown in our paper, blood transfusion represents 90.3% (1118/1238) of all bleeding events in kidney transplant recipients (2614/2991 (87.4%) for percutaneous native kidney biopsies). For this reason, although we do believe that a sensitivity analysis is a good idea, it is not possible to perform it. We regret it, and we hope that the reviewer will understand our reason.

We propose to add in the Discussion section :

To better assess the specific role of anemia as a risk factor of bleeding it would have been interesting to perform a sensitivity analysis using nephrectomy, hemorrhage/hematoma and nephrectomy as bleeding events (excluding therefore blood transfusion). Unfortunately, blood transfusion represented 90.3% of all bleeding events in kidney transplant recipients (and 87.4% for percutaneous native kidney biopsies), and therefore such an analysis was not possible.

3.Authors describe a significant center effect. However, the use of blood transfusion after bleeding maybe different among centres and different practices may explain current findings.

We agree.

We propose to add in the Discussion section :

We observed a center effect regarding the rate of bleeding after biopsy in kidney transplant recipients. The use of blood transfusion after bleeding maybe different among centres and different practices may explain current findings.

4.If the sensitivity analysis would suggest that anemia is the real driver of the association with the endpoint as defined by authors, the reviewer would suggest changing the title to suggest the need for blood transfusion and the discussion to highlight that the focus is not the entity of the bleeding per se but the need for blood transfusion the scope of the analysis. Ultimately, blood transfusion is what we should avoid to reduce the risk of allo-sentitization 

As indicated, such an analysis was not feasible.

5.Please add in the text a Breuer (larger ?) explanation of how the new bleeding risk is calculated.

To create the score points, the regression coefficients with p<0.01 were divided by the smallest coefficient and rounded to the nearest integer, which assign a given number of points for each significant predictor in the mortality model. [1,2].

1. Lloyd-Jones DM, Wang TJ, Leip EP, Larson MG, Levy D, Vasan RS, D'Agostino RB, Massaro JM, Beiser A, Wolf PA, Benjamin EJ (2004) Lifetime risk for development of atrial fibrillation: the Framingham Heart Study. Circulation 110 (9):1042-1046. doi:10.1161/01.cir.0000140263.20897.42
2. Mehta HB, Mehta V, Girman CJ, Adhikari D, Johnson ML (2016) Regression coefficient-based scoring system should be used to assign weights to the risk index. J Clin Epidemiol 79:22-28. doi:10.1016/j.jclinepi.2016.03.031

6.Also explain how the AUC analysis are carried out (bleeding risk as a continuous variable? What is the cut off? As dichotomous variable high vs low risk (what is the definition of high vs low risk)?

AUC were carried out with the score as a continuous variable. We thus did not consider a specific cut-off for a high vs a low risk in this analysis. 

7.Can authors compare the additive prognostic value of the new score when compared to other risks scores?

Although several studies identified risk factors for bleeding (mainly after native kidney biopsies), we are not aware of any other risk score (beside the one we published in Halimi et al, (CJASN 2020, ref : 24). The new score we propose is not superior to the older one but is easier to calculate, as indicated in the Discussion Section.